# Advancing Nigerian Indigenous Poultry Health and Production, Use of Probiotics as Viable Alternatives to Antibiotics: A Review

**DOI:** 10.3390/antibiotics14080846

**Published:** 2025-08-21

**Authors:** Shedrach Benjamin Pewan, Dennis Kabantiyok, Paulinus Ekene Emennaa, Joshua Shehu Dawurung, Christiana J. Dawurung, Reuben Kefas Duwil, Olufunke Olufunmilola Olorundare, Hassan Yader Ngukat, Moses Gani Umaru, Garba Mathias Ugwuoke, Chuka Ezema

**Affiliations:** 1Veterinary Extension Division, National Veterinary Research Institute NVRI, Vom 930010, Nigeria; 2Central Diagnostic Laboratory, National Veterinary Research Institute NVRI, Vom 930010, Nigeriaemeonepaulie@gmail.com (P.E.E.); 3School of Chemistry & Molecular Biosciences, University of Queensland, Brisbane 4072, Australia; j.dawurung@student.uq.edu.au; 4Department of Pharmacology and Toxicology, Faculty of Veterinary Medicine, University of Jos, Jos 930105, Nigeria; dawurungc@unijos.edu.ng; 5Department of Animal Production, Faculty of Agriculture, University of Jos, Jos 930105, Nigeria; reubenduwil@gmail.com; 6Forest Research Institute of Nigeria, Federal College of Forestry, Jos 930105, Nigeria; olufunmilolaolorundare@gmail.com; 7Department of Animal Health & Production Technology, Adamawa State College of Agriculture, Ganye 641113, Nigeria; hassanngukat76@gmail.com; 8Department of Animal Health & Production, Federal Polytechnic, Bali 672102, Taraba State, Nigeria; ganimoses57@gmail.com; 9Department of Physiology and Pharmacology, Faculty of Veterinary Medicine, University of Nigeria, Nsukka 410001, Nigeria; matthias.ugwuoke@unn.edu.ng; 10Department of Animal Health and Production, Faculty of Veterinary Medicine, University of Nigeria, Nsukka 410001, Nigeria; chuka.ezema@unn.edu.ng

**Keywords:** antibiotic growth promoters, probiotics, microbiome, immunity, antimicrobial resistance

## Abstract

Poultry is a vital component of global meat production, with particular importance in Nigeria and Africa, as it promotes food security, economic growth, and rural livelihoods. Indigenous chickens, although less productive, are well adapted to local environments and provide significant socio-economic and nutritional benefits. The rising demand for animal protein and concerns over antimicrobial resistance (AMR) necessitate the development of sustainable alternatives to antibiotics in poultry production. Probiotics have emerged as effective feed additives that enhance gut health, immunity, nutrient absorption, and overall productivity. While extensively studied in commercial poultry, research on probiotics in Nigerian Indigenous Ecotype Chickens (NIECs) remains limited. Key challenges in indigenous poultry systems include low productivity, disease vulnerability, limited veterinary access, and environmental pressures. Addressing these requires improved management practices, infrastructure, veterinary support, and enabling policies. Multi-strain probiotics, particularly those containing *Lactobacillus*, *Bifidobacterium*, and *Bacillus* species, demonstrate promise in enhancing productivity, improving product quality, promoting environmental sustainability, and ensuring food safety. Focused research on local probiotic strains, field trials, farmer education, and policy support is crucial for harnessing the full benefits of probiotics and transforming indigenous poultry farming into a resilient and sustainable sector.

## 1. Introduction

Poultry is the world’s most consistently consumed meat globally, valued for its cost-effectiveness, efficient production, short generation interval, and superior nutritive value. It is a form of high-quality protein and a source of the most essential amino acids and essential micronutrients, such as B vitamins and selenium. It has been demonstrated to improve muscle growth, immunity, and metabolic health, especially amid increasing demand for protein-based diets [1]. It is easier to manage and elicit low capital and operating costs than other types of livestock [2]. In Nigeria and Africa, animal husbandry is indispensable for human health, women’s economic empowerment, and the resilience and environmental sustainability of rural communities [3].

With a growing population, the global demand for slaughtering and animal products, such as meat, milk, and eggs, is increasing, posing significant challenges to the livestock industry in keeping pace with this growing demand. The world’s human population is growing significantly and is projected to reach 9.7 billion by 2050 [4]. Improving production efficiency in poultry and livestock is crucial for sustainably feeding an ever-growing population while also protecting animal and human welfare by reducing antibiotic use and promoting environmentally friendly and ethical farming practices [5]. The population of poultry in Africa by 2022 was approximately 2.4 billion, with Egypt leading at 300 million and Nigeria second at 249 million [6]. Hence, Nigeria, a major player in the industry, has a large poultry population, with the sector ranging from commercial to small-scale farm holdings, including indigenous and exotic breeds, and is distributed across various regions of the country.

The indigenous chickens are also referred to by various names in small-scale village, free-range, local, scavenging, family, or backyard production systems [7]. They are mostly unimproved and are characterized by low productivity but are majorly beneficial to rural society because of their hardiness and good adaptation to harsh environments and require little veterinary health care [7,8]. Although indigenous chickens are not selected for increased productivity, they are endowed with traits that render them well-adapted to harsh tropical environments. Native chickens consume a free-range, natural diet consisting of house scrapings, grazing grasses, seeds, insects, and nuts, supplemented with additional commercial feed [9]. They also have small eggs and a tenacious instinct to sit on and hatch them unaided. They are also highly resistant to diseases of local endemicity [10].

According to Kasimanickam et al. [11], administering antibiotics to poultry at sub-therapeutic doses promotes growth, and this often leads to the emergence of antibiotic-resistant bacteria. Another drawback to this practice of administering antibiotics at subtherapeutic levels is the presence of drug residues in eggs and meat, which allows bacterial organisms to develop resistance to these antibiotics [12]. More often, withdrawal periods are not observed to avoid antibiotic residues in the meat and eggs that humans ingest, as they do not apply to meat-producing animals and their byproducts, which are meant for human consumption [13]. Due to their scavenging behavior, indigenous birds often encounter waste, sewage, and other contaminated materials, thereby increasing their susceptibility to infections caused by resistant bacteria, such as *E. coli* and *Campylobacter* [14]. Conversely, commercial birds are reared in controlled settings that implement biosecurity measures and vaccination protocols, which mitigate pathogen exposure; this environment allows for more precise administration of antibiotics under strict veterinary supervision [15].

The use of antibiotic growth promoters (AGPs) has been prohibited in most countries due to the growing problem of antibiotic resistance and antibiotic residues in poultry and livestock products, which pose a risk to human health. Antibiotics are also prohibited in most parts of the globe for use in animal husbandry at subtherapeutic levels. Their use is limited to cases prescribed by a veterinary surgeon for the treatment of confirmed clinical cases. As the impending threat of AMR deepens, scientists must intensify their search for novel alternatives to traditional antimicrobials to maintain performance and protect the efficacy of existing therapies while acquiring justifiable solutions within a rapidly diminishing collection of practicable choices [16]. Diverse alternatives to AGPs have been tested as feed additives in poultry, yielding mixed results. In recent years, an array of innovative options for antibiotics, such as probiotics [17], prebiotics [18], postbiotics [19], phytochemicals [20], bacteriophages [21], bacteriocins [22], exogenous enzymes [23], vaccines, and antimicrobial peptides [24], have been developed as part of efforts to address AMR and AMU. Most promote growth, improve livestock productivity, and support animal and human health.

Probiotics, also known as direct-fed microbials (DFMs). They have attracted significant global interest in improving poultry health and performance, presenting a feasible alternative to antibiotic growth promoters in response to the growing demand for antibiotic-free and organic poultry. They are also gaining general recognition in the medical field for their varied health benefits and attaining notable success across sectors such as agriculture, biotechnology, and pharmaceuticals, owing to their preventive and therapeutic properties [25,26,27]. Integrating probiotics into animal diets has, therefore, increased growth, production efficiency, and disease resistance (immunity). It sustains the gut microbiota, which plays a critical function in poultry and livestock health and productivity. It does so by promoting beneficial bacteria, inhibiting pathogenic species, and enhancing more efficient nutrient digestion and absorption, host metabolism, and health in general [28,29,30].

Probiotics enhance nutrient digestion and absorption by augmenting the synthesis and action of digestive enzymes such as lipase, protease, and amylase. Probiotics are well-known for their wide range of health benefits, which largely depend on the bacterial strain, dose, and delivery volume, chicken genetics, inoculation site, concentration, dosage, and duration of administration [31,32]. There is a paucity of information on the use of probiotics in indigenous chickens in Nigeria. The objective of this review is to evaluate the potential of probiotics as sustainable alternatives to antibiotics in enhancing the health and productivity of Nigerian indigenous poultry. It is essential to conduct an appraisal using current literature and assess instances where probiotics have been utilized as supplements in chickens, particularly their impact on growth promotion and production performance, strengthening the gut microbiome and immune system, and improving meat quality. This should be performed while also considering the potential use in decreasing ammonia and greenhouse gas emissions.

## 2. An Overview of the Indigenous Poultry Industry in Nigeria

Indigenous poultry rearing presents exceptional opportunities for smallholder farmers, including women and children, to generate a profitable income and diversify their livelihoods, thereby contributing to poverty alleviation and economic empowerment. It is fundamental for food and nutrition security, as poultry products supply animal protein and essential micronutrients [33]. This type of farming is often more environmentally friendly than large commercial systems because it utilizes local resources, minimal infrastructure, and local breeds that thrive in harsh climates [34]. Local chicken meat is also regarded for its organoleptic properties, such as excellent taste, desirable flavor, and juicy tenderness [35]. It also boasts an impressive nutritional profile, rich in healthy protein and essential vitamins and minerals that support overall health. Compared to other traditional meat sources, indigenous chicken has a significantly lower percentage of saturated fat and cholesterol and a higher rate of unsaturated fatty acids, including oleic acid (a monounsaturated fatty acid, or MUFA) and total polyunsaturated fatty acids (PUFAs), which may be considered a heart-healthy option for health-conscious consumers [36].

Despite their multiple benefits, indigenous chickens are faced with numerous challenges by their keepers, which are attributed to the extensive management systems offered to them compared to exotic chicken breeds. However, malnutrition, inadequate extension services, and a lack of technical farming skills have been implicated in poor growth and development [37]. Bird keepers commonly keep these birds in small living quarters near them; this promotes exposure to various zoonoses, such as salmonellosis, chronic respiratory disease, and lousiness. On the other hand, such systems are also challenged by inadequate housing and food. Idiosyncrasies such as temperature fluctuations and extreme weather exacerbate these factors, which in turn influence the health and performance of these birds. Moreover, their relatively small body size, the number of flocks, and age variation within the flock decrease the overall level of meat production, and they have lower egg production, making them inefficient from a commercial standpoint [38]. Furthermore, disease control is one of the main challenges facing the poultry sector in Africa, specifically in smallholder production systems, as escalating production costs and a lack of sufficient vaccines limit farmers [39]. Free-range chickens are more prone to infections, such as Newcastle disease, coccidiosis, and fowl pox, disease vectors that distribute avian hemiparasites, resulting in increased morbidity and mortality risk. A low level of biosecurity, insufficient drugs, and the fact that smallholder farmers have little technical knowledge of how to prevent and control diseases further exacerbate the problem [40,41]. Vaccination programs in these systems are difficult to implement. Challenges include limited access to veterinary support services, difficulties in maintaining a cold chain to preserve vaccine efficacy, and inconsistent production and distribution of vaccines, particularly in resource-constrained countries [42]. Although indigenous chicken production practices could be sustainable, these challenges need to be addressed through innovative interventions, such as improved feeding and management practices, adopting the best biosecurity measures, and developing climate change adaptation technologies. These interventions are needed to contribute to increased productivity and income.

### Antimicrobial Resistance

Antimicrobial resistance (AMR) has emerged as one of the most critical global public health emergencies, driven by the overuse and misuse of antimicrobial agents. This has led to a decrease in their capacity to tackle infections successfully. This top global health challenge is threatening the lives of millions of livestock and humans. Indigenous birds typically necessitate a higher quantity of antibiotics for the treatment of bacterial infections compared to commercially raised birds; this disparity is attributed mainly to their increased exposure to environmental pathogens and antibiotic-resistant strains. There are reports of free-ranging indigenous birds exhibiting AMR without being treated with antibiotics [43,44]. The detection of resistant genes may have resulted from environmental contamination. Antimicrobials, also known as antibiotics, have been recognized for decades for their ability to boost feed conversion efficiency, promote growth, prevent and treat animal diseases, and enhance productivity, thereby keeping pace with the demand for poultry meat and eggs. These antimicrobials are often administered in sub-therapeutic doses through animal feed, also to improve feed efficiency, leading to more significant mass gain and higher product yields [45,46]. This fosters a balanced gut microbiome and strengthens the immune system [47,48]. Collectively, these gains improve poultry’s overall health, productivity, and profitability. The use of these drugs in animal husbandry dates to the mid-1940s, when the surge in antibiotic production for treating wounded soldiers during World War II made these medications widely available. This accessibility enabled their successful integration into veterinary practices, yielding extraordinary results [49]. However, the gains of employing antibiotics are becoming less clear as concerns grow over their prolonged and inappropriate use as feed supplements, resulting in hazards of drug residues in animal products [50] and the development of antibiotic-resistant bacteria. For example, traces of antibiotics have been found in food items, raising potential health risks for consumers [51]. The detection of antibiotic residues in poultry meat and eggs poses substantial dangers to human health, including antibiotic sensitivities, allergic reactions, disturbances to the gut microbiota, increased bacterial resistance, and significant potential losses within the food chain [52]. Furthermore, long-term antibiotic use in poultry and livestock can disrupt the stability of intestinal microbiota, eventually impairing weight gain, survival rates, and feed efficiency [53,54].

The growing dissemination of antimicrobial-resistant bacteria from animals to humans, primarily through the consumption of animal products or direct contact, poses a substantial clinical risk due to the transmission of resistance genes. The increase in multidrug-resistant pathogens results from the inappropriate use of antibiotics in both veterinary and human medicine [55,56]. In addition, the use of antibiotics in food animals can lead to other food safety hazards, such as an increased incidence of allergic reactions, development of cancer, and carcinogenic and teratogenic outcomes in neonates and treatment failures in humans [57,58,59]. The growing challenge of antimicrobial resistance, resulting from indiscriminate or excessive use, is more prevalent in developing countries and poses considerable public health risks [60]. Resistant bacteria make simple infections harder to treat. This often leads to the need for more expensive alternatives if they are available, resulting in more devastating complications and resultant death. These drawbacks have led several countries worldwide to ban antibiotics at subtherapeutic doses in poultry and other food animals [61]. The prohibition on the use of antibiotics in poultry and livestock has led to increased morbidity and mortality, reduced feed intake and efficiency, and a resultant decrease in body weight. As a result of its use in human and veterinary practice and its consequent environmental contamination, AMR has now become the basis of the One Health Initiative, intricately linking the fields of agriculture, environment, and public health [62]. Consequently, an integrated global approach within these segments is required to tackle and effectively handle this growing menace. Across African countries, there is a prevalent misuse of antibiotics, inadequate antibiotic regulations, and a dearth of regional monitoring of AMR and antimicrobial usage (AMU) [63,64].

## 3. Biotics

In the gut health and nutrition perspective, “biotics”, a collective noun, represents a class of supplements including prebiotics, probiotics, synbiotics, and postbiotics. These compounds promote the development and activity of beneficial gut bacteria, and as a result, these substances have lately been the core of vast research as prospective antibiotic substitutes in poultry and livestock production [65].

### 3.1. Prebiotics

In recent years, prebiotics have garnered considerable attention due to their numerous benefits in enhancing the health of poultry and livestock. Prebiotics are non-digestible nutrients, specifically fermentable oligosaccharides (2–10 monosaccharide units), that have been demonstrated to favorably manipulate the composition and fermentation patterns of the gastrointestinal microbiota, facilitating the establishment of a healthy microbiota and regulating the host’s immunity and metabolism [66,67]. Notable prebiotics include mannan oligosaccharide (MOS), galactooligosaccharides (GOSs), fructooligosaccharides (FOSs), and alginate oligosaccharide (AOS) [68,69]. Morgan [70] stated the requisites for qualifying prebiotics to include that they should not be metabolized in the upper gastrointestinal tract (GIT), they must operate as a selective nutrient for valuable microorganisms in the gastrointestinal system, and they should initiate physiological reactions that offer benefits to the host. Fermentation products of prebiotics include short-chain fatty acids, principally volatile fatty acids (acetate, propionate, and butyrate) and lactic acid, which offer energy to the host and reduce gastrointestinal pH, thereby hindering the multiplication of acid-sensitive bacteria [71,72]. Therefore, integrating locally sourced prebiotic ingredients into poultry feed provides a sustainable and welfare-oriented approach to intensive farming. This strategy enhances animal performance and health outcomes, offering a cost-efficient and environmentally responsible alternative that advances the goal of a more resilient and ethical poultry industry. GIT is a complex, tube-like structure common to almost all species, with the small intestine functioning as a vital region accountable for the major digestion and assimilation of dietary nutrients [73].

### 3.2. Synbiotics

Synbiotics, which blend probiotics and prebiotics, deliver beneficial bacteria along with the nutrients they need; this combination helps enhance the survival and establishment of these helpful microbes in the gut [74]. A study by Hossain and his colleagues [75] confirmed that the effectiveness of synbiotics depends on various interconnected factors, including the exact dosages administered, routes of delivery, the specific combinations of prebiotic and probiotic strains, and the probiotic species used. They also stated that the breed and strain of the poultry, diet, location, levels of exposure to physiological stress, stocking densities, and ambient environmental temperatures impact the efficiency of synbiotics. Synbiotics, a synergistic combination of live probiotics and selectively utilized prebiotics, can be administered in ovo to poultry, enhancing gut colonization by beneficial bacteria and thereby improving digestion, health, and protection against pathogens [76].

Research on synbiotics has centered mainly on broiler and other commercial strains such as Arbor Acres and Ross 308 within broad poultry production contexts [77,78]. Although a few Nigerian reviews touch on probiotics, prebiotics, and postbiotics in the national poultry sector, no report has conducted direct comparative trials on synbiotics in indigenous breeds [79]. Notably, work on native Mandarah chicks in Egypt has tested different synbiotic administration routes, assessing outcomes in growth, gut morphology, and immune function [80]. While these findings come from outside Nigeria, the approach offers meaningful insight. Targeted trials on the country’s own indigenous ecotypes remain a pressing need.

### 3.3. Postbiotics

Postbiotics, also known as pharmacobiotics, metabiotics, or heat-killed probiotics, are an emerging area of research and are gaining fast attention as they boost poultry and livestock health and production. They refer to metabolic byproducts or constituents obtained from probiotic bacteria that are less susceptible to various environmental factors, including temperature, pH, and processing methods [80], and they confer a spectrum of health and nutritional benefits to the host.

Supplementing diets with postbiotics has numerous benefits for the host, as it improves the architecture of intestinal villi, lowers the populations of Enterobacteriaceae, increases the fecal pH, and increases the number of lactic acid bacteria, leading to enhanced health and growth performance in broilers [81]. Chaney et al. [82] reported that incorporating *Saccharomyces cerevisiae* fermentation products in poultry feed might be a reliable approach for reducing the incidence of *Salmonella enterica* and improving food safety and overall poultry health. The intestinal microbiota produces postbiotics, such as Gamma-Aminobutyric Acid (GABA) from L-glutamic acid, short-chain fatty acids from carbohydrates, indole from amino acids, and polyphenolic acids from dietary sources; these compounds are crucial for maintaining our bodies’ balance and health [83,84]. These metabolites are products formed when gut bacteria ferment dietary components, playing a fundamental role in how the host’s body interacts with the microbiome. For example, short-chain fatty acids (SCFAs) are synthesized by the host gut microbiota and are key players in managing insulin and glucose levels, inflammation, fat tissue development, and even certain cell growth [85]. This underscores their significance in various physiological processes. Some postbiotics have direct antimicrobial activities through the reinforcement of the gut barrier, competition for specific receptors required by pathogens, modulation of host gene expression, or even pH changes in the environment [86]. In other words, postbiotics are more stable, safer, fast-acting, and durable than probiotics [87]. Due to their resistance and effectiveness, they are a desirable option for various reasons, regardless of the presence of live bacteria.

Recent literature reveals only one Nigeria-based investigation into postbiotic and paraprobiotic supplementation, focusing solely on broiler chickens [88]. The study measured immune responses in commercially farmed flocks, offering no insight into the country’s indigenous breeds. This gap underscores the importance of conducting robust, randomized trials that explore how postbiotics influence native ecotypes such as the Fulani, Ogun, and Yoruba, where breed-specific responses may hold valuable implications for poultry health and productivity.

### 3.4. Probiotics

The term “probiotic,” derived from the Greek word meaning “for life,” signifies non-pathogenic, live microorganisms (usually a culture of a single strain, or a mixture of different strains) that, when ingested in sufficient amounts (10^6^–10^7^ cfu/g), confer health benefits to the host via a symbiotic affiliation [89]. Probiotics are live microorganisms, typically bacteria and yeasts, that offer numerous health benefits to the host organism when consumed in sufficient amounts through food or supplements [90]. This represents a complete departure from the long-held misconception that all microorganisms are inherently harmful to livestock and humans. Over the years, the functional food sector has experienced a substantial increase in the promotion of probiotic-based products, primarily driven by growing consumer interest in dietary choices that offer enhanced physiological benefits [91]. Over the past decade and a half, probiotics have garnered significant attention for their therapeutic potential and have been extensively employed in treating a broad spectrum of pathological disorders. Probiotics have been associated with a wide range of health-enhancing effects, including but not limited to the following: enhancing lactose digestion for lactose-intolerant patients [92], regulating various types of cancer, especially colorectal cancer [93], and lowering blood cholesterol levels [94]. Additional benefits include reduction of blood glucose and cholesterol levels, reduction of obesity and hypertension, and resistance to allergens [95].

Dietary supplements, including probiotics, have become essential in poultry production. They stimulate optimal growth, efficient nutrient digestion and absorption, boost gut integrity, bolster immunity, and sustain a healthy balance of gut microbiota [96,97]. While the addition of probiotics to feed remains the most typical method of administration on poultry farms, a range of other delivery systems is also available; these include oral gavages (administered as drops or vaccines), aerosol sprays, granules, tablets, coated capsules, and sachets enclosing powdered preparations [98].

The sustenance of high feed efficiency is critical for the poultry industry’s ability to fulfill the increasing global demand for poultry products. However, this is a significant challenge, as intensive bird production is highly susceptible to infectious disease outbreaks, particularly in areas facing shifting climatic conditions that aggravate vulnerability to health-related disruptions [97]. The high cost of feed ingredients has consistently driven the price of livestock feeds to account for approximately 70% of the total cost of poultry production. Therefore, any measure that will reduce the price or quantity of this farm input would always be welcomed by farmers and make farming more sustainable. Optimizing conversion of feed to body weight is a critical determinant of commercial success in livestock production (there is a close relationship between profitability and efficiency of conversion of feed by the animal into live weight gain).

Current innovations have shown that probiotics promote human health and enhance food safety by eliminating harmful microorganisms and their toxins in food. The association between diet and gut microbiota has emerged as a significant focus for promoting animal and human health, with probiotic supplements gaining increasing acceptance as an efficient means of influencing the gut microbiome [99,100].

It has long been recognized that the variation in gut microbiota exerts a significant influence on host metabolism, nutrient digestion, growth performance, and the welfare of the host. Over the past few years, probiotics have been considered a safe and effective alternative to growth-promoting antibiotics in poultry production, attracting significant scientific and commercial attention due to their benefits in enhancing gut health and production performance, as well as combating antibiotic resistance in poultry [101]. Additionally, probiotics can react differently depending on the strain used and are susceptible to degradation when the temperature is unstable, making their effects unpredictable.

#### 3.4.1. Characteristics of a Good Probiotic

An effective probiotic strain must be non-pathogenic, survive particularly under acidic and bile-rich conditions, have an elevated number of viable cells, demonstrate metabolic activity, and have a positive impact on the host’s health. The ability to adhere to and colonize the mucosa of the digestive tract, improving the functions of the intestinal tract and retaining stability and viability throughout storage and field application [102,103]. A microbial strain can be deemed a probiotic if it meets specific, stringent safety and efficacy standards, comprising genetic stability, acid and bile resistance, gut adhesion, non-existence of pathogenicity, anti-genotoxic capacity, and lactic acid production [104]. It must also meet rigorous processing requirements, promote microbiota diversity, and have a reasonably brief reproductive cycle.

The mechanisms of action of probiotics are multifactorial and not fully characterized. However, they support the body by releasing antimicrobial compounds, competing with harmful microbes for space on the gut lining, reinforcing the intestinal barrier, and fine-tuning the immune system’s responses. Probiotics exert their beneficial effects through several mechanisms, including the competitive exclusion of pathogens from adhesion sites, the manufacture of antimicrobial substances, the improvement of the intestinal mucosal barrier, the modulation of the immune system, and the production of neurotransmitters [76]. Recent findings corroborate that probiotic survival in the GIT is sometimes compromised by factors such as acidity, bile salts, oxygen, and processing stress [105,106]. This preceded the development of synbiotics (a mixture of probiotics and prebiotics) to improve microbial viability and efficacy and hamper the growth and multiplication of pathogenic species.

Probiotics’ advantages stem from their distinct structural traits and ecological roles. Their presence strengthens the intestinal barrier, supports a diverse microbiota, and helps prevent the growth of harmful pathogens.

This review attempts to comprehensively investigate the potential roles of probiotics to improve food safety by eliminating bacterial, fungal, viral, and parasitic pathogens and neutralizing their toxins from meat and eggs. Furthermore, recent cases of probiotic commensals’ antimicrobial activity in the body condition after ingesting contaminated foods have also been summarized. These data show that different probiotics can inactivate pathogens and neutralize or detoxify various biological agents in food present in the host body following ingestion.

The most used probiotics are *Bacillus subtilis*, *Enterococcus faecium*, *Lactobacillus acidophilus*, *Bacillus amyloliquefaciens*, *Bacillus licheniformis*, *Lactobacillus plantarum*, *Lactobacillus casei*, *Lactobacillus rhamnosus*, *Lactococcus lactis*, *Pediococcus acidilactici*, *Bacillus coagulans*, *Bifidobacterium bifidum*, *Carnobacterium divergens*, *Lactobacillus delbrueckii*, *Lactobacillus farciminis*, *Lactobacillus fermentum*, *Lactobacillus paracasei*, and *Streptococcus salivarius* [107].

#### 3.4.2. Role of Probiotics in Augmenting a Balanced Gut Microbiome and the Immune System

With the expansion of gut microbiota research, its pivotal influence on poultry health and growth is becoming increasingly evident, indicating that this microflora is an essential modulator in physiological development and the overall condition of poultry production systems. Recent studies have reported that optimal intestinal health in birds requires a balanced gut microflora to maintain a continuous interface with its host and harmonious coexistence [108]. A normal and healthy gut ecosystem of poultry is essential for its general health, nutrient digestion and absorption, immune system, and overall welfare performance. The microbiome encompasses a wide range of abundant microbiota (trillions of microorganisms) that have colonized a host and their bioactive products, nucleic acids, proteins, lipids, polysaccharides, and metabolites, which are formed through interactions with the host and affected by the environmental factors [109]. It is well established that gut microbiota diversity performs pivotal functions in host metabolism. It digests nutrients for growth, promotes gut microbiota diversity, eliminates pathogenic bacteria, enhances the intestinal barrier, maintains the immune system, and maintains the overall health of the host [31,110,111]. Any disruption of these could lead to health problems, which probiotics can help decipher. Enhancing gut health is essential for chickens to reach their genetic potential and achieve a low feed conversion ratio (FCR). Any compromise, mainly caused by enteric diseases, not only reduces performance but also has a direct and adverse effect on overall profitability [112]. Several advances can be made concerning preventing contaminated food by various pathogenic and toxigenic microorganisms. Numerous investigations have demonstrated that multiple probiotics, in addition to exerting beneficial effects on the health of hosts, can destroy or neutralize foodborne pathogens and their toxins in foods, thus improving food safety. Additionally, the differences in geography, age, sex, and feeding habits during the chicken’s rearing influence the composition of the microbial population in the chicken’s intestine, with significant potential effects on poultry health and productivity [113].

By regulating gut permeability through tight junctions, probiotics enhance the gut health of poultry. The tight junction protein complex controls ion movement, prevents pathogenic bacteria from passing between cells, and maintains intestinal epithelial integrity [114]. However, alterations in these proteins could lead to damage to the mucosal barrier, resulting in the translocation of pathogens within the intestinal epithelium. Maintaining the intestinal epithelium’s integrity is crucial in poultry, as it serves as a selective barrier to limit the access of pathogenic organisms and toxins whilst allowing efficient nutrient absorption and utilization [115]. When disrupted due to factors such as enteric disease or inflammation caused by stress, this barrier can be associated with enhanced intestinal permeability, microbiota imbalance, and systemic inflammation, all of which have negative implications for growth, feed efficiency, and productivity [116,117]. In this sense, probiotics can help support or improve gut barrier function through various mechanisms, including the prevention of pathogenic bacteria and modulation of tight junction proteins. Thus, stabilized tight junction proteins shore up the epithelial barrier, protecting the host from pathogen invasion into internal organs. This has resulted in comprehensive improvements in gut permeability and the suppression of pathogenic bacteria [112]. Probiotics play a crucial role in maintaining a balance in intestinal flora. They accomplish this by inhibiting the proliferation of detrimental bacteria and promoting the growth of beneficial microbes. This dual effect not only supports the intestinal barrier but also stimulates the immune response, resulting in superior growth performance and overall well-being in the animals. When provided at sufficient levels, probiotics can help establish a healthier microbial balance in the poultry gut, thereby improving digestion and supporting overall health, and providing a “shield” against harmful bacteria and pathogens [118]. Immunoglobulin G (IgG), the most abundant serum antibody comprising over 75% of blood antibodies, and secretory immunoglobulin A (sIgA), produced in the lamina propria, together play vital roles in systemic immunity and mucosal defense in the respiratory and gastrointestinal tracts [119,120]. This defensive layer efficiently blocks pathogens from attaching to cell surfaces, thereby significantly reducing the likelihood of infection and microbial invasion [121]. Ensuring a robust immune function is vital for the healthy development of chickens under all management systems.

Recent in vivo results have emphasized the efficiency of *Lactobacillus* administration in remarkably reducing *Salmonella* colonization in a variety of species, including poultry, pigs, and mice, demonstrating that *Lactobacillus* administration is a biologic control method against enteric bacterial infections in avian and livestock [122,123]. The chicken gut microbiota represents a dynamic microbial community that is influenced by the age of the host, method of administration and route of infectious exposure, genotype, dietary inputs, and husbandry system. These factors collectively play roles in the stability and functionality of the gut ecosystem during the animal’s life [92,124,125]. The enteric microbes inhabiting the GIT play a crucial role in nutrient and drug metabolism, detoxification, and prevention of pathogenic colonization, but they also contribute to the induction and regulation of significant aspects of the host’s innate and adaptive immune response [126,127]. The gut microbiota, through interaction with the gut-associated lymphoid tissue (GALT), has a predominant influence on the development of the avian immune system. Probiotics boost both innate and adaptive defenses from the microflora by affecting the immune system, thereby enhancing resistance against infectious agents of the host [128].

GIT employs a combination of physical and immune defenses to confine its diverse microbial community to the gut lumen. The mucosal layers act as the principal endogenous defense system, capturing potential infectious agents and readily clearing them from the body through passage through the intestine [129]. Simultaneously, these mucus structures provide adhesion sites and metabolic substrates that allow bacteria to colonize and survive [130]. Disturbances in the gut microbiota (also known as dysbiosis) or its interaction with the immune system may adversely affect the health of the intestine and may arouse autoimmune diseases, which also highlights the significant role of the immune system in homeostasis of the host–microbe interaction and also the importance of keeping a microbial balance for immune homeostasis and the host’s general well-being [131,132]. Alterations of the gut microbiota can be disrupted by antibiotic use, changes in diet, ageing, and infections, which can lead to potential discrepancies in pathogenic flora, inflammatory reactions, and metabolic diseases [131]. Therefore, interventions to maintain healthy microbiota are necessary to enhance chickens’ immune response, health, and productivity. However, little is known about the microbiota of the Nigerian indigenous chickens. Nigerian indigenous chickens may differ from those of imported and exotic chickens in terms of their microbial composition and functional profiles, which could contribute to their edibility and potentially harbor unique probiotic strains with health benefits for the host. Therefore, there is a need to determine the gut microbiota composition of indigenous Nigerian chickens and establish reference data for this purpose. This limits our ability to design probiotic interventions that target specific microbial imbalances or health challenges (Figure 1) [133].

#### 3.4.3. Role of Probiotics in Promoting Growth and Production Performance

Numerous studies have highlighted the beneficial roles of probiotics in improving growth performance and overall health status in laying birds and broilers. Incorporating probiotics into poultry diets has been demonstrated to increase feed conversion efficiency, feed quality, nutrient availability, boost body weight gain, and increase overall productivity, possibly attributed to improved nutrient digestibility [134,135]. This may be due to enhancing beneficial microbial populations, curbing pathogenic species, or bolstering intestinal barrier integrity, leading to a significant improvement of the host’s health and resilience [136,137]. Multi-strain probiotic formulations may offer improved health benefits not only through their collective actions but also due to the precise ratio of each strain, enabling synergistic interactions such as enhanced gut adherence and pathogen suppression, while delivering diverse protective effects through various mechanisms of action [138,139].

Hens in the late laying stage, which accounts for approximately 50% of the production cycle, undergo physiological changes, including lowered nutrient absorption, reduced immune function, and disrupted lipid metabolism, collectively decreasing laying and egg quality [140]. This results in considerable economic losses for poultry farmers. Eggshell quality is one of the critical determinants of egg shelf life and marketability, with shell damage during handling, transport, and storage accounting for 6–10% of global losses, substantially affecting profitability [141]. Albumen height is a key marker of internal egg quality, indicating the freshness and general state of the egg white [142]. By modifying the gut microbiome, probiotics enhance nutrient bioavailability, enabling laying hens to efficiently meet the metabolic demands of eggshell formation, particularly during peak production levels [143]. Carcass yield, along with breast and drumstick ratios, in part indicates broiler meat productivity and acts as an indirect measure of their overall growth productivity [144]. This leads to increased productivity, improved animal welfare, enhanced environmental sustainability, and greater farmer profitability. There is a synergy between intestinal microstructure and microbes that influences digestion and nutrient absorption, thereby regulating body weight and feed intake [145]. Villus height (VH) and crypt depth (CD) are essential markers of gut function; taller villi and shallower crypts signify improved nutrient absorption capacity and greater epithelial renewal, with less energy spent on cell production [146,147]. This ultimately sustains improved growth performance via an enlarged surface area for efficient nutrient uptake.

Recent histomorphometry analysis has confirmed that single or multiple probiotic supplementations often lead to an improved structure of the small intestine by increasing the villus height to crypt depth (VH:CD) ratio, boosting goblet cell density, and advancing the synthesis of Muc2, a fundamental constituent of the mucus layer [31]. Therefore, supplementing diets with probiotics considerably enhanced intestinal histomorphology in chickens and other livestock species. This is likely due to the probiotics’ role in promoting villi cell repair, maturation, and metabolic activity. This aids in improved gut architecture and possibly enhanced nutrient absorption and overall well-being.

Zeng and co-researchers [148] documented that supplementing broiler feed with a blend of *Clostridium butyricum*, *Bacillus subtilis*, and *B. licheniformis* increased total and mean broiler body weight. Zou and his colleagues stated that carcass yield, along with proportions of breast and drumstick meat, partially shows broiler meat and growth performance [144]. Moreover, the authors demonstrated that supplementing diets with a 0.025% probiotic combination of *Bacillus subtilis*, *Clostridium butyricum*, and *Enterococcus faecalis* remarkably improved breast muscle fraction over the control group. Probiotic supplementation, therefore, leads to an improvement in feed conversion ratio (FCR) in poultry, suggesting that birds require less feed to attain target growth or egg production levels. This enhancement directly leads to a decrease in production costs and better economic returns for poultry operations [149]. Multi-strain combinations and behavior/welfare implications (e.g., stress mitigation) have not been explored in Nigerian breeds, even though recent global reviews have accentuated such effects.

#### 3.4.4. The Role of Probiotics in Reducing Ammonia, Hydrogen Sulphide, and Greenhouse Gas Emissions

High-density poultry production not only leads to the generation of odorous volatile organic compounds and noxious gases (ammonia (NH_3_), hydrogen sulphide (H_2_S), and mercaptan) but also greenhouse gases such as methane (CH_4_) and carbon dioxide (CO_2_). Unwanted smells may occur with excessive NH_3_ and H_2_S gases. Odor generated from poultry and livestock waste decomposition is often associated with neighborhood complaints and social concerns; recently, the use of probiotics and microbial products has been used for odor alleviation associated with poultry and livestock management [150,151] (Figure 2).

In an in vitro study, Mi and his colleagues [153] indicated that the use of a microbial consortium, i.e., *Pichia guilliermondii* + *Bacillus subtilis* + *Lactobacillus plantarum* in a 1:2:1 ratio, resulted in reductions in NH_3_ gas emissions by 46%, as revealed by its potential for mitigating the environmental impact in poultry production systems. Probiotic application led to a decrease in crude protein digestibility, pH, ammonium nitrogen, and valerate, as well as a reduction of urease and uricase activity. It also led to an increase in urea, purine trione, nitrate nitrogen, total volatile fatty acids, and acetate, indicating remarkable changes in nitrogen utilization and microbial fermentation [153]. Some recent research observed a reduction of fecal *E. coli* and *Salmonella*, increased *Lactobacillus*, a suppression of NH_3_ and H_2_S emissions in broiler feces because of probiotic supplementation, and a good connection between the amount of pathogenic bacteria and gas emissions in an experimental group of birds. A matter that hitherto gave support to some previous reports on associative relations between suppression of odor and microbial balance and nutrient utilization [100,154]. When human beings and other creatures encounter these gases, they cause pain and agitation, suppress appetite, and can also lead to chronic respiratory diseases. The addition of probiotics to poultry feed has been linked to a reduction in NH_3_ and H_2_S gas emissions. As probiotics in food improve digestion and nutrient absorption, fewer nutrients are available in manure for microbial action and decomposition [155].

#### 3.4.5. Meat Quality

Meat quality is characterized by elements including lean to fat ratio as well as palatability features including visual appearance, odor, drip loss, color, texture, pH (acidity), water-holding capacity, fatty acid composition, tenderness, flavor, and juiciness [156,157]. Including probiotics in the diet is one of the strategic options for managing and improving the quality of meat from chickens, particularly broiler chickens. It serves to reduce the use of other additives and ingredients that affect animal health. At the same time, they contribute feed ingredients with good nutritional value, increasing the value of poultry products [18]. Several responses [158] were reported, including the effects of probiotics on carcass traits, such as fat deposition, meat quantity and quality, and some of these could be seen in Table 1 and Figure 3. The addition of probiotics to the diets of poultry animals also improves the nutritional quality of poultry meat and reduces the chances of chronic diseases, such as obesity, cardiovascular diseases, and type 2 diabetes [159].

Iakubchak et al. [175] found improvement in carcass associated with muscle accretion due to probiotics, resulting in better formation of muscle tissue and, consequently, better growth performance. According to a study by Ivanovic [176], treatment by some probiotics in the diets of 42-day-old Arbour Acres broilers in comparison to control led to a higher moisture level and lower fat and protein content of drumstick meat. This indicates a possible difference in the composition of the meat, provoked by the probiotic supplementation. All researchers found that meat from animals supplemented with probiotics was less moist and had a higher crude protein content, which is probably due to an increase in protein synthesis. High moisture levels are usually associated with poor meat quality; thus, this change in the meat composition suggests an upgrade in the overall quality of the FS products [177,178,179].

#### 3.4.6. Impact of Probiotics on Stress Management in Poultry

Livestock and poultry are continuously exposed to a variety of environmental factors, including high stocking density, poor handling and transportation, disease outbreaks, feed changes and restrictions, high ammonia buildup, and temperature extremes. These have a profound impact on the profile and integrity of the gut microbiome [180] and subsequently affect the health, performance, and overall productivity of chickens. The absence of sweat glands, the presence of thermally insulating plumage, and a relatively high body mass-to-surface area ratio render poultry particularly susceptible to a range of environmental stressors [181].

Emerging research suggests that as global temperatures continue to rise, the negative impact of oxidative stress on the health and performance of poultry and livestock will become increasingly pronounced [182]. Studies have established a link between heat-induced oxidative damage and impaired growth metrics in avian species [183,184]. Probiotics are effective interventions for reducing stress in poultry by rebalancing the gut microbiota, modulating the hypothalamic–pituitary–adrenal (HPA) axis, and regulating the microbiota–gut–brain axis. Heat stress in chickens triggers the HPA axis and increases corticosterone concentrations, which may compromise intestinal health [78]. Probiotics are known to modulate the composition of the gut microbiota by regulating pathogenic bacteria and promoting the growth of beneficial bacteria, thereby contributing to the improvement of intestinal integrity. This will result in better gut health, as well as increased resistance to stress-induced disruptions. Probiotics may influence the HPA axis by reducing systemic levels of circulating inflammatory cytokines and microbial antigens, thereby guarding against neuroinflammation [185]. This balance modulates the stress response in chickens, promoting overall chicken health. The microbiota–gut–brain axis is an ancient bidirectional signaling system that connects the gut microbiota and the central nervous system. Probiotics modulate this axis through metabolites, including short-chain fatty acids, which serve as an energy source for intestinal cells, and exert anti-inflammatory effects in the intestinal tract, as well as influence neurotransmitters [186], potentially affecting mood and behavior. This interaction could facilitate a decrease in stress-like behaviors and an improvement in welfare in poultry. The use of probiotics in poultry feeding has been reported to alleviate negative behaviors associated with stress, such as aggression and feather pecking, and to enhance productivity. *Bacillus subtilis* addition resulted in lower levels of aggressive behavior, as observed in laying hens subjected to a social challenge [187]. Hence, probiotic supplementation in feed leads to improved animal welfare, reduced stress, and better productivity in poultry and other animals.

## 4. Challenges, Specific Knowledge Gaps, and Future Research

The increasing risk of antimicrobial resistance and the growing demand for safe, residue-free animal products have driven attention toward the potential use of probiotics as alternatives to AGPs in the poultry industry. This is also the case for Nigerian indigenous poultry, which is a significant contributor to the livelihoods and food security of rural and semi-urban households.

The potential of probiotics to improve growth performance, feed utilization, and production in poultry has been extensively explored. In indigenous breeds, they ensure the integrity of the intestine by stimulating the balanced growth of the intestinal microflora, improving nutrient absorption and intestinal immunoregulation. For instance, some studies have reported that including probiotics in the diet of chickens can lead to an increase in carcass yield and meat quality, resulting from reduced fat deposition and improved muscle texture and taste characteristics, which are increasingly sought after by modern consumers. However, these should be investigated in Nigerian local chickens.

In the context of environmental concerns, organic acids act as probiotics and can help mitigate the emission of noxious gases such as ammonia and hydrogen sulphide associated with poultry waste. Probiotics, by enhancing gut fermentation processes, decrease nitrogenous excretion, thereby reducing the production of greenhouse gases and contributing to the sustainability of poultry production systems.

Notwithstanding the encouraging advantages, several challenges limit the acceptance of probiotics in Nigeria. Among them is the significant lack of indigenous poultry gut ecology-adapted strains. Furthermore, there are limited locally based breed-specific studies on the efficacy and safety of commercial probiotic strains. Poor farmer knowledge, a lack of generic standards governing regulatory conduct, and restricted access to affordable and quality-controlled products are additional constraints that prohibit the practical application.

Further studies are needed to focus on the isolation and characterization of indigenous probiotic strains, which can adapt promptly to local poultry genotypes and conditions. Multi-local field studies on performance, gut microbiota modulation, and eco-environmental impact are needed under both semi-intensive and extensive husbandry conditions. Integrated extension strategies for disseminating recommendations and promoting farmers’ adoption also need to be developed and strengthened.

## 5. Conclusions

The Nigerian Indigenous Ecotype Chicken (NIEC) plays a vital role in rural food systems, livelihoods, and cultural traditions. Though they exhibit lower productivity compared to commercial breeds, their resilience to local environmental stressors and disease makes them indispensable to sustainable poultry systems in Nigeria. However, the growing demand for animal protein, coupled with rising concerns about antimicrobial resistance (AMR), highlights the urgent need for viable, sustainable alternatives to antibiotics.

Probiotics have emerged as promising feed additives that enhance gut health, immunity, and nutrient utilization, contributing to improved performance and food safety. Yet, while extensive research has focused on their use in commercial poultry, application in NIEC remains limited. To ensure effective adoption, probiotic strategies must be tailored to the genetic and microbial profiles of indigenous breeds.

Key priorities moving forward include the isolation and characterization of probiotic strains native to the NIEC gastrointestinal tract, which could offer more context-specific benefits. Additionally, cost-subsidized field deployment of validated probiotic products is crucial to encourage widespread adoption among smallholder farmers. Government policy support is equally vital, particularly in the form of funding for local research, regulation of probiotic products, and integration of probiotics into national livestock health strategies.

An integrated, locally driven approach that combines scientific research, practical field application, farmer education, and institutional backing will be essential to unlocking probiotics’ full potential in enhancing Nigeria’s indigenous poultry sector’s health, resilience, and productivity.

## Figures and Tables

**Figure 1 antibiotics-14-00846-f001:**
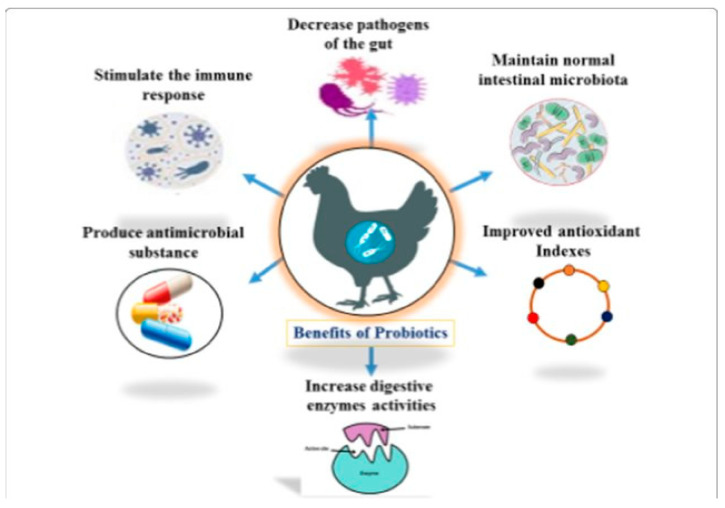
Illustrates the multifaceted benefits of probiotics in poultry. These include reducing gut pathogens, maintaining intestinal microbiota, improving antioxidant indexes, enhancing digestive enzyme activity, producing antimicrobial substances, and stimulating immune responses. Collectively, probiotics support better gut health, immunity, digestion, and oxidative balance, contributing to overall poultry health and performance. Source: Rajput et al. [133].

**Figure 2 antibiotics-14-00846-f002:**
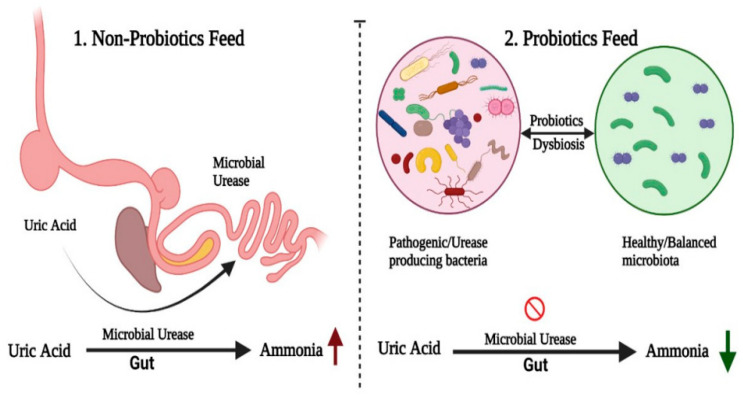
Comparison of non-probiotic and probiotic poultry feeds on gut microbiota and ammonia production. Non-probiotic feeding (**left**) increases gut urease activity, where pathogenic bacteria metabolize uric acid into excess ammonia, resulting in elevated levels (red upward arrow). In contrast, probiotic feeding (**right**) suppresses urease-producing bacteria, restores a balanced microbiota, and lowers ammonia generation (green downward arrow). This shift not only improves gut health but also reduces environmental ammonia load. Source: Khalid et al. [152].

**Figure 3 antibiotics-14-00846-f003:**
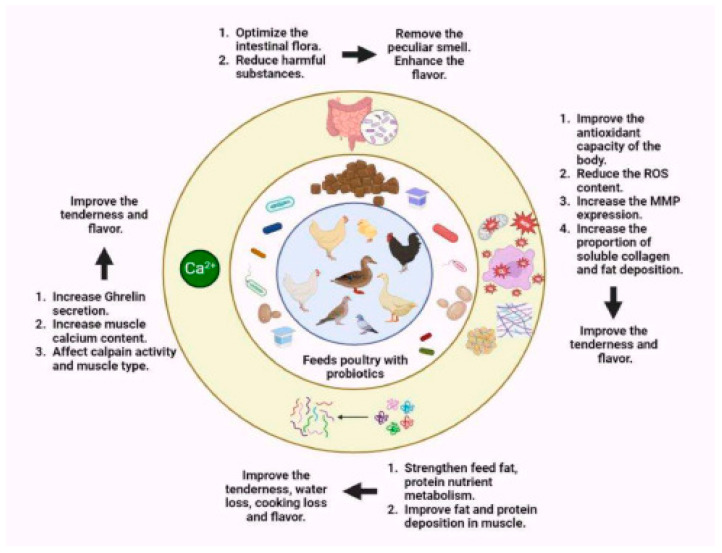
Illustrates how feeding poultry with probiotics enhances meat quality. Probiotics optimize gut flora, reduce harmful substances, boost antioxidant capacity, regulate calcium, and improve nutrient metabolism. These actions increase muscle calcium, collagen, and fat deposition, reduce reactive oxygen species (ROS), and enhance ghrelin secretion. Overall, probiotics improve tenderness, water retention, and flavor, and reduce cooking loss in poultry meat. Source: Dong et al. [158].

**Table 1 antibiotics-14-00846-t001:** Comparison of selected studies on the impact of probiotics on poultry performance parameters.

Probiotic Strains	Health/Performance Effects	Reference
*Lactobacillus reuteri*	Early supplementation improves microbial diversity, increases beneficial microbes, reduces pathogen levels, and lowers disease susceptibility (up to 6 weeks of age).	[148,160]
*Bacillus subtilis, Clostridium butyricum*, *Enterococcus faecalis*	Supplementation increases the thymus and bursa of Fabricius relative to body weight (by 29.3%), indicating enhanced humoral and cellular immune responses and improved overall immunity.	[100,161]
Multi-strain probiotics (unspecified)	Enhance immune responses, increase lysozyme activity, and boost IgA, IgG, IgM, and T-lymphocyte activity in intestinal mucosa.	[99,162]
*Bifidobacterium* spp., *Lactobacillus casei* (0.5%)	Improve intestinal mucosa, inhibit harmful bacteria from spreading, and significantly increase egg production.	[163]
*Bacillus subtilis*	Increased egg production, improvements in egg weight, shell thickness, and Haugh unit, increased egg freshness and quality	[164]
*Bacillus subtilis*, *Enterococcus faecium*	Enhance feed intake and conversion efficiency; improve growth and egg production.	[165]
*Bacillus subtilis*, yeast cell wall components	Improve egg weight, shell thickness, yolk pigmentation, albumen height, and Haugh unit.	[166]
*Clostridium butyricum*, *B. subtilis*	Improve the villus height to crypt depth ratio; better nutrient absorption.	[167]
*Enterococcus lactis*	Enhances nutrient absorption, growth, egg production, feed conversion, and calcium/phosphorus metabolism (notably during late laying).	[168,169]
*Lactobacillus rhamnosus*	Improves egg-laying performance, feed-to-egg ratio, and eggshell strength.	[170]
*Pediococcus acidilactici*	Maintains laying rate and feed efficiency when the metabolizable energy in the diet is reduced.	[140]
*Saccharomyces cerevisiae*	An increase in hen-day egg performance and a reduction in circulating cholesterol, mitigate oxidative stress.	[171]
Multi-strain preparation probiotics	Improved body weight gain, feed conversion ratio, reduced cholesterol, increased HDL content, lowered LDL, lower total triglycerides, meat yield, Increased meat protein in broilers	[172]
*Bacillus*-based probiotics	Decreased abdominal fat and improved fatty acid composition with increased unsaturated fatty acids in broilers.	[173]
*Bacillus*-based probiotics	Improved hen-day egg production and average egg weight, especially at higher dosages.	[174]

## Data Availability

As this is a literature review, this statement is not applicable.

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
