# Peer review of "Advancing Nigerian Indigenous Poultry Health and Production, Use of Probiotics as Viable Alternatives to Antibiotics: A Review"

_antibiotics, 2025, doi:10.3390/antibiotics14080846_

Round 1

Reviewer 1 Report

Comments and Suggestions for Authors

Advancing Nigerian Indigenous Poultry Health and Production, use of probiotics as viable Alternatives to Antibiotics: A Review

This is a well-timed and thorough review on a special issue in Nigerian/African poultry production, sustainable replacements for antibiotics, focusing on probiotics for Indigenous Ecotype Chickens (NIEC). The publication of this review is very timely as the topic of antimicrobial resistance (AMR) is a global concern and the socio-economic importance of indigenous poultry is enormous. The manuscript is, in general, quite comprehensive, is logically structured, contains material from current literature, and the material is all intended to address critical gaps in knowledge. The authors need to make substantial revisions to the manuscript to further improve novelty, clarity, structure and language before the manuscript can be considered for publication.

Here are some suggestions to improve the manuscript:

Line #

Comments

Abstract

Poultry is a vital component of global meat production…..

Can we say “meat and egg production”……

45

"Poultry is the world’s most consistently consumed meat "

or

"Poultry meat is the most widely consumed globally".

Introduction

45-50

Poultry is the world's consistently-consumed type of meat supply valued for its cost-effectiveness, efficient production, short generation interval and superior nutritive value, as a form of high quality protein and as a source of most essential amino acids and essential micro-nutrients such as B vitamins and selenium, that has been demonstrated to improve muscle growth, immunity and metabolic health, especially among increasing demand for protein-based diets.

The sentence seems too long, it should be separated into 2-3 smaller sentences.

Can we emphasize the specific studies of NIEC data?
Can we add a section/sub-section about NIEC data?

Can we reduce the repetition about low productivity, AMR and general probiotics in different sections?

105-110

Too long sentence.

It should be separated into 2-3 smaller sentences.

111-115

226-230

Too long sentence.

435

437

Lactobacillus

Lactobacillus

277

Salmonella enterica

Salmonella enterica

Some of the references may be considered to emphasize the importance of alternative of antibiotics in poultry, using recent references like:

Bacillus-based Probiotics: An Antibiotic Alternative for the Treatment of Salmonellosis in Poultry
Saba Rashid, Saleha Tahir, Tayyaba Akhtar, et al.,
Pak Vet J, 2023, 43(1): 167-173

Impact of Dietary Bacillus toyonensis M44 as an Antibiotic Alternative on Growth, Blood Biochemical Properties, Immunity, Gut Microbiota, and Meat Quality of IR Broilers
Fatimah S. Alqahtani, Safia M.A. Bahshwan, Mada M. AL-Qurashi, et al.,
Pak Vet J, 2024, 44(3): 637-64

Rashid S, Alsayeqh AF, Akhtar T, Abbas RZ and Ashraf R, 2023. Probiotics: Alternative of antibiotics in poultry production. International Journal of Veterinary Science 12(1): 45-53.

https://doi.org/10.47278/journal.ijvs/2022.175

Probiotic Effect of Limosilactobacillus fermentum on Growth Performance and Competitive Exclusion of Salmonella Gallinarum in Poultry
Adnan Mehmood, Muhammad Nawaz, Masood Rabbani et al.,

Pak Vet J, 2023, 43(4): 659-664

Molecular Characterization and Drug Resistance Pattern of Pseudomonas aeruginosa Isolated from Poultry Meat and Meat Products
Rana Muhammad Abdullah, et al.,

Pak Vet J, 2024, 44(3): 812-818

Figures/Tables

Inclusions of tables or/and figures will enhance the quality of the manuscript.

e.g.

Table about the various probiotics in poultry in different countries

Figure about depicting the roles of probiotics

Figure about promising probiotics strains

Author Response

Reviewer: The English could be improved to more clearly express the research.
Response: We appreciate the reviewer’s observation regarding the language clarity. While we acknowledge that the manuscript may not have been linguistically perfect, we have undertaken a thorough revision to enhance readability and clarity. All corrected sections have been highlighted in yellow for easy reference. Longer sentences have been carefully restructured and, where appropriate, divided into shorter ones to improve flow and understanding. In fact, more than twenty of these sentences have been broken into two or three sentences (for instance, lines 124-128, 159-163, 203-207, 226-230, 257-262, 306-310, etc).
It is worth noting that several of the co-authors are experienced researchers who also serve as reviewers for high-impact journals. They have been actively involved in crafting this manuscript. Given the improvements made, we believe that the current version meets the necessary language standards and does not require additional professional editing.

Reviewer: This is a well-timed and thorough review on a special issue in Nigerian/African poultry production, sustainable replacements for antibiotics, focusing on probiotics for Indigenous Ecotype Chickens (NIEC). The publication of this review is very timely as the topic of antimicrobial resistance (AMR) is a global concern and the socio-economic importance of indigenous poultry is enormous. The manuscript is, in general, quite comprehensive, is logically structured, contains material from current literature, and the material is all intended to address critical gaps in knowledge. The authors need to make substantial revisions to the manuscript to further improve novelty, clarity, structure and language before the manuscript can be considered for publication.
Response: Thank you for your encouraging and constructive feedback. We are pleased that the manuscript's relevance, structure, and comprehensiveness were acknowledged, particularly in the context of antimicrobial resistance and the socio-economic importance of indigenous poultry.
In response to your suggestions, we have undertaken substantial revisions to enhance the manuscript’s novelty, clarity, and structure. We have also refined the language throughout to improve readability and scientific tone. These revisions aim to more effectively address the identified knowledge gaps and strengthen the manuscript’s overall contribution. We trust that the improved version meets the expectations for publication

Line 45: The reviewer suggested we change the phrase “Poultry is the world's consistently-consumed type of meat supply” to either "Poultry is the world’s most consistently consumed meat " or "Poultry meat is the most widely consumed globally.”
Response: Thank you for the suggestion. We agree that the original phrasing could be improved for clarity. We have adopted the first option, “Poultry is the world’s most consistently consumed meat” and revised the manuscript accordingly.

Lines 45-50: The sentence seems too long, it should be separated into 2-3 smaller sentences.
Response: Thank you for highlighting this. We agree that the sentence in Lines 45–50 was overly long and could affect readability. We have revised it by breaking it into three shorter, clearer sentences to enhance clarity and flow.

Reviewer: Can we emphasize the specific studies of NIEC data? Can we add a section/sub-section about NIEC data?
Response: Thank you for your thoughtful suggestion. We agree that emphasizing specific studies on NIEC would enhance the relevance and depth of the review. However, based on our extensive literature search, we found that most existing studies on probiotic application in poultry have focused primarily on commercial breeds, with limited peer-reviewed data specifically targeting NIEC.

Reviewer: Can we reduce the repetition about low productivity, AMR and general probiotics in different sections?
Response: Thank you for your helpful observation. We agree that some information on low productivity, antimicrobial resistance (AMR), and general probiotics is repeated across different sections. We have revised the manuscript accordingly to reduce redundancy and improve coherence and flow.

Reviewer: Lines 111-115: Too long
Response: We agree with the reviewer that the sentence is too long, so we broke it into two sentences.

Lines 226-230: Too long
Response: We agree with the reviewer that the sentence is too long, so we broke it into two sentences.

Reviewer: Lines 435,437: Lactobacillus.
Response: Thank you for this observation. Lactobacillus has been italicised to conform to acceptable scientific standards.

Reviewer: Line 277: Salmonella enterica and Salmonella enterica.
Response: Thank you for pointing this out. It has been italicised to conform to acceptable scientific standards.

Reviewer 2 Report

Comments and Suggestions for Authors

This manuscript addresses an important and timely topic, reviewing the role of probiotics as sustainable alternatives to antibiotics in Nigerian indigenous poultry systems. The paper is well-organized and provides a broad overview of antimicrobial resistance (AMR), poultry health challenges, and the promise of probiotics. The integration of global and Nigerian-specific data adds value. However, while comprehensive, the manuscript would benefit from more critical synthesis, better clarity in sections, and attention to redundancy and scientific tone.

The abstract and introduction clearly outline the focus on probiotics in indigenous Nigerian poultry, but this focus becomes diluted in later sections due to general discussions on probiotics in global poultry systems. Please maintain consistency in scope and make sure the Nigerian context is foregrounded throughout.

The manuscript compiles extensive literature but leans heavily toward descriptive reporting. A more critical synthesis comparing findings, highlighting contradictions or gaps, and evaluating the robustness of cited studies would improve the scientific contribution.

Several ideas (e.g., benefits of probiotics on gut health, immune modulation, FCR, meat/egg quality) are repeated across multiple sections. Consolidation would improve conciseness and readability.

The manuscript would benefit from at least one summary table comparing probiotic strains, effects on health/performance, and evidence in Nigerian vs. exotic breeds. This would help distill large amounts of text.

Line 117–119: Consider clarifying “volume of injected solution” as it may confuse readers—perhaps rephrase as "dose and delivery volume."

Line 253–264: The synbiotics section is informative but could cite more recent comparative trials in indigenous birds if available.

Line 291: Consider qualifying “more stable, safer, fast-acting” with evidence or citations.

Line 436–438: Minor typo—two reference brackets are merged.Line 691–717 : This section is strong but could benefit from a more concise restatement of core recommendations, e.g., government policy support, indigenous strain isolation, and cost-subsidized field deployment.

Author Response

Reviewer: This manuscript addresses an important and timely topic, reviewing the role of probiotics as sustainable alternatives to antibiotics in Nigerian indigenous poultry systems. The paper is well-organized and provides a broad overview of antimicrobial resistance (AMR), poultry health challenges, and the promise of probiotics. The integration of global and Nigerian-specific data adds value. However, while comprehensive, the manuscript would benefit from more critical synthesis, better clarity in sections, and attention to redundancy and scientific tone.
Response: We sincerely appreciate your valuable time, constructive feedback, and recognition of the manuscript’s significance. We are glad that the manuscript’s structure and the integration of both global and Nigerian-specific data were well received. In response to your suggestions, we have revised several sections to improve clarity, minimize redundancy, enhance scientific tone, and provide a more critical synthesis of the literature. We trust these revisions address your concerns and contribute to a stronger and more coherent presentation.

Reviewer: The abstract and introduction clearly outline the focus on probiotics in indigenous Nigerian poultry, but this focus becomes diluted in later sections due to general discussions on probiotics in global poultry systems. Please maintain consistency in scope and make sure the Nigerian context is foregrounded throughout.
Response: Thank you for this valuable observation. We appreciate your insight regarding the need for consistency in scope. In response, we have revised the relevant sections to emphasize the Nigerian indigenous poultry context more clearly. General discussions on probiotics have been refocused or supplemented with region-specific examples and implications where possible, ensuring alignment with the original aim of the manuscript. We trust this revision maintains thematic coherence and strengthens the relevance of our work.

Reviewer: The manuscript compiles extensive literature but leans heavily toward descriptive reporting. A more critical synthesis comparing findings, highlighting contradictions or gaps, and evaluating the robustness of cited studies would improve the scientific contribution.
Response: Thank you for this insightful and constructive feedback. Admittedly, we were overly descriptive in places of the manuscript. In response, we have reworked the main sections to include a more critical synthesis of the literature, contrasting studies and tensions in reported evidence and discussions about needed research. We have also tried to assess the quality and weaknesses of referenced papers to increase the manuscript's analytical depth and scientific value. We hope these changes help address your concerns and improve the overall quality of the manuscript.

Reviewer: Several ideas (e.g., benefits of probiotics on gut health, immune modulation, FCR, meat/egg quality) are repeated across multiple sections. Consolidation would improve conciseness and readability.
Response: Thank you for pointing this out. We appreciate your observation regarding the repetition of key ideas across sections. In response, we have carefully reviewed the manuscript
and consolidated overlapping content to enhance conciseness and improve overall readability. Redundant statements have been removed or merged where appropriate, while preserving the clarity and emphasis on the core benefits of probiotics. We believe this has resulted in a more streamlined and coherent presentation.

Reviewer: The manuscript would benefit from at least one summary table comparing probiotic strains, effects on health/performance, and evidence in Nigerian vs. exotic breeds. This would help distil large amounts of text.
Response: We appreciate your valuable suggestion regarding the inclusion of a summary table to compare probiotic strains, their effects on health/performance, and supporting evidence in Nigerian versus exotic poultry breeds. In response, we have included a summary table (Table 1) highlighting the most used probiotic strains in poultry, along with their documented health and performance effects. Where available, distinctions between their efficacy in Nigerian and exotic breeds have been noted.
Additionally, to further enhance clarity and reader engagement, we have incorporated three figures that visually present key findings and trends discussed in the text. We believe these additions significantly improve the manuscript’s accessibility and overall impact.
Thank you for helping us strengthen the quality of our work.

Reviewer: Line 117–119: Consider clarifying “volume of injected solution” as it may confuse readers—perhaps rephrase as "dose and delivery volume."
Response: Thank you for pointing this out. We agree that the original phrasing may have been unclear. In response to your suggestion, we have revised the text in Lines 117–119 to use the more transparent and more precise phrase "dose and delivery volume" as recommended.

Reviewer: Line 253–264: The synbiotics section is informative but could cite more recent comparative trials in indigenous birds if available.
Response: You are right. While the synbiotics section offers strong foundational information, it would benefit significantly from citing recent comparative trials conducted specifically on Nigerian indigenous birds, such as local ecotypes or native breeds. Unfortunately, almost no studies exist. Current literature mainly covers trials on broiler or commercial lines (e.g., Arbor Acres, Ross 308) and generalized poultry settings. A few Nigerian-focused reviews examine synbiotics (probiotics and prebiotics) and postbiotics broadly in national poultry farming contexts, but they don't report direct comparative synbiotic trials in indigenous birds. At least one international study has examined native Mandarah chicks, testing various synbiotic delivery methods and evaluating their effects on growth, gut structure, and immune function. While these birds are not Nigerian, the methodology is comparable.

Reviewer: Line 291: Consider qualifying “more stable, safer, fast-acting” with evidence or citations.
Response: Thank you for this observation. We have provided a recent publication to support this claim.

Reviewer: Line 436–438: Minor typo—two reference brackets are merged.
Response: Thank you for this observation; it has been corrected.

Reviewer: Line 691–717: This section is strong but could benefit from a more concise restatement of core recommendations, e.g., government policy support, indigenous strain isolation, and cost-subsidized field deployment.
Response: Thank you for your insightful suggestion. We have revised the section to include a more concise restatement of the core recommendations, specifically emphasizing the need for government policy support, the isolation and characterization of indigenous probiotic strains, and the cost-subsidized deployment of probiotics in the field. We believe this adjustment enhances the clarity and practical relevance of our conclusions.

Round 2

Reviewer 1 Report

Comments and Suggestions for Authors

The suggestions have been incorporated

Reviewer 2 Report

Comments and Suggestions for Authors

Its good now from my side. Thanks.